**Multiple scattering correction factor of quartz filters and the effect of filtering particles**
**mixed in water: implications to analyses of light-absorption in snow samples**
Jonas Svensson[1*], Johan Ström[2], and Aki Virkkula[1]
[1]Atmospheric Composition Research, Finnish Meteorological Institute, Helsinki, Finland
[2]Department of Environmental Science and Analytical Chemistry, Stockholm University,
Stockholm, Sweden
*Now at Institute for Geosciences and Environmental Research, University of Grenoble
Alpes, Grenoble, France
*Correspondence to* J. Svensson (jonas.svensson@fmi.fi)
**Abstract**
The deposition of light-absorbing aerosols (LAA) onto snow initiates processes that lead to increased
snowmelt. Measurements of LAA, such as black carbon (BC) and mineral dust, have been observed
globally to darken snow. Several measurement techniques of LAA in snow collects the particulates on
filters for analysis. Here we investigate micro-quartz filters optical response to BC experiments where
the particles initially are suspended in air or in a liquid. With particle soot absorption photometers
(PSAP) we observed a 20% scattering enhancement for quartz filters compared to the standard PSAP
Pallflex filters. The multiple-scattering correction factor ($C_{ref}$) of the quartz filters for airborne soot
aerosol is estimated to ~3.4. In the next stage correction factors were determined for BC particles mixed
in water and also for BC particles both mixed in water and further treated in an ultrasonic bath.
Comparison of BC collected from airborne particles with BC mixed in water filters indicated
approximately a factor of two higher mass absorption cross section for the liquid based filters, probably
due to the BC particles penetrating deeper in the filter matrix. The ultrasonic bath increased absorption
still further, roughly by a factor of 1.5 compared to only mixing in water. Application of the correction
functions to earlier published field data from the Himalaya and Finnish Lapland yielded MAC values
of ~7 – 10 $m^2$ $g^{-1}$ at $\lambda$ = 550 nm which is in the range of published MAC of airborne BC aerosol.

1     **Introduction**
Soot refers to carbonaceous particles formed during the incomplete combustion of hydrocarbon fuels,
and includes black carbon (BC) and organic carbon (OC), but can also include other elements, such as
sulfates. ~~consists of black carbon (BC) and organic carbon (OC) particles formed during the incomplete~~
~~combustion of carbonaceous fuels~~. As the most light-absorbing aerosol (LAA) by unit per mass, BC is

36 highly efficient in absorbing solar radiation, and is a vital component in Earth's radiative balance (Bond

37 et al., 2013). Once the particles are scavenged from the atmosphere, possibly far from their emission

38 source, BC can reach a snow surface and decrease the snow reflectivity (Warren and Wiscombe, 1980;

39 Flanner et al., 2007). This will lead to accelerated and increased snowmelt, observed in different snow

40 environments across the globe (see e.g. recent review by Skiles et al., 2018). Perhaps most notably is

41 High Mountain Asia and its extensive cryosphere, where large emission sources of LAA in close

42 proximity is affecting the region's snow and ice (e.g. Xu et al., 2009; Gertler et al., 2016; Zhang et al.,

43 2017).

44

45 There are a variety of methods for measuring BC ~~Measurements of BC consists of a variety of methods~~,

46 which is reflected in BC being operationally defined. A common practice is to measure the change in

47 transmission of a filter collecting aerosol. The measured signal (i.e. optical depth of the filter) is

48 thereafter applied with correction factors to generate atmospheric concentrations of so-called equivalent

49 black carbon (eBC) according to the BC nomenclature (Petzold et al., 2013). The correction factors

50 account for: 1) the loading of aerosol on the filter since the detection signal decreases with increased

51 aerosol content; 2) the multiple scattering of light that is enhanced in the filter substrate; 3) and the

52 enhancement from the deposition of other light scattering aerosol. One instrument used for light

53 absorption measurements is the Particle Soot Absorption Photometer (PSAP), utilizing Pallflex filters.

54 As an alternative for the optical filter analysis of eBC, another approach is to apply the thermal-optical

55 method (TOM), providing organic carbon (OC) and elemental carbon (EC) mass of the aerosol on the

56 filter. With this method, EC refers to the carbon content of carbonaceous matter (Petzold et al., 2013),

57 and can be assumed to be the main light-absorbing element of BC. The technique involves a stepwise

58 heating procedure, therefore creating a need to use micro quartz fiber filters. These filters have been

59 used in numerous studies with filtering snow and ice samples, and thereafter analyzed to determine the

60 EC and OC content of the samples (e.g. Hagler et al., 2007; Forsström et al., 2009; Meinander et al.,

61 2013; Ruppel et al., 2014; Zhang et al., 2017). In Svensson et al. (2018), measurements with TOM were

62 combined with an additional transmittance measurement to further ~~characterize the LAA present on the~~

63 ~~filter samples~~ investigate the relative contribution from BC and other LAA particles present in snow

64 samples. The study involved laboratory tests, as well as comparisons to ambient snow samples taken

65 from different environmental settings. One lesson from this study was that the optical properties of

66 absorbing particles on quartz filters must be better understood. In particular when using melted snow

67 samples.

68

69 The overarching goal of this paper is to further investigate micro quartz fiber filters optical behavior

70 when sampling BC particles in a liquid (to simulate snow sampling). An advantage of using these filter

71 is that the sample can be analysed readily using TOM to arrive to an EC concentration on the filter

72 (where MAC values are not needed). Thi~~se~~ aim is pursued through a series of laboratory studies ~~of BC~~

~~filter deposition in an airborne phase, as well as when the same BC particles are mixed in water and~~
~~filtered onto the quartz filters (to simulate snow sampling).~~ Our approach is to compare the use of quartz
fibre filter for air and liquid samples to the much better characterized pallflex type filer used in
commercial PSAPs. Hence, we are not intending to determine a universal MAC value, but rather to
understand differences in the observations that might be due to the filter substrate or handling of the
sample. We do not intend to answer all possible issues with filter sampling, but will concentrate on the
difference using the two filter types in air samples, the difference between air and liquid samples with
respect to the quartz fibre filter, and finally the potential effect from treating the liquid samples using
ultrasound.

## 2      Materials, instruments, and data analyses

### 2.1      Soot aerosol production and sampling

A schematic picture of the experiment is presented in Fig. 1, and the methods used in each step are
outlined in the subsections (2.1.1 and 2.1.2) below, as well as the instrumentation used (sections under
2.2). Section 2.3 explains the data processing. The soot used consisted of particles collected by chimney
cleaners in Helsinki, Finland, and this particular soot batch is from small-scale oil-based burning. The
same soot has been applied in different experiments previously (Peltoniemi et al., 2015; Svensson et
al., 2016; 2018).

### 2.1.1 Airborne sampling

Soot aerosol were sampled onto filters in an airborne phase and as a part of liquid solution. In the
airborne aerosol tests, soot was blown into a cylindrical experimental chamber (0.8 m height × 0.45 m
diameter) through a stainless steel tube (25 mm outer diameter) consisting of a y-shaped bend of 130°,
creating a size-separation of the aerosol. Essentially a virtual impactor, this set-up allowed the smaller
sized particles to continue with the airflow into the chamber, while the larger (and heavier) particles
were deposited into a waste pipe through inertial separation (see section 2.2.3 for further description
and results in section 3.1.1.).~~The soot used consisted of particles collected by chimney cleaners in~~
~~Helsinki, Finland, and this particular soot batch is from small-scale oil-based burning. The same soot~~
~~has been applied in experiments previously (Peltoniemi et al., 2015; Svensson et al., 2016; Svensson et~~
~~al., 2018).~~ From the experimental chamber a sample inlet (copper, 6 mm outer diameter) simultaneously
fed two PSAPs and a portable aerosol spectrometer (Grimm 1.108). One of the PSAPs had quartz fiber
filter punches mounted, while the other had standard PSAP filters installed. This set-up was alternated
among the PSAPs in between the experimental runs during the experiment, to have both PSAPs assessed
with the different filters. In total, 22 different experimental rounds were made with various amounts of
aerosol deposited to the substrates.

### 2.1.2 Liquid sampling

In the liquid experiments, the same soot batch and procedure were used as above, but the outlet pipe was submerged into a 20 L container filled with deionized, purified Milli-Q (MQ) water. From this liquid solution, different small amounts (between 10-100 mL) were extracted and mixed with additional MQ water to further dilute the sample (to a typical total volume of 400 mL). This was performed to get a range of filters with different EC concentrations and optical depths. The total number of liquid-generated filters was 35. Some selected liquid samples (n=10) were exposed to an ultrasonic bath (for at least 15 mins) prior to filtration to further mix the soot solutions. All of the liquid solutions were filtered onto the same quartz filters used in the airborne test, applying the same filtering principles and analysis procedures as used previously (Svensson et al., 2018). Punches from dried filters had their transmittance first measured using a PSAP, followed by EC concentration measurements (TOM). This procedure was also applied to the quartz filters from the airborne experiment.

121

## 2.2    Instruments

### 2.2.1    Absorption measurements

Absorption was measured with two Radiance Research 3-wavelength PSAPs (S/N 90 and S/N 100) at $\lambda$=467 nm, 530 nm, and 660 nm (Virkkula et al., 2005). One of them was loaded with Pallflex E70-2075W filter that is generally used with the instrument, while the other was loaded with micro quartz fiber filters (Munktell, grade T293). The flows were calibrated with a Gilian Gilibrator bubble flow meter and set to 0.5 LPM. Higher flow rates were not used here since the quartz filter tends to be more fragile and may not withstand higher flows. The sample spot diameters of the PSAPs were measured with an Eschenbach scale loupe with a 0.1 mm graduation ten times each. The average diameters ($\pm$ standard deviation) were $5.04 \pm 0.10$ mm and $5.05 \pm 0.10$ mm, giving corresponding spot areas of $19.9 \pm 1.6$ mm$^2$ and $20.0 \pm 1.6$ mm$^2$. The aim was to use identical face velocities, i.e. average velocity of aerosol perpendicular to the filter (e.g. Müller et al., 2014) through both filter materials. The essentially identical spot areas meant also that we had tuned the flow rates identical. In addition, to study whether the PSAPs themselves affect the results we used both filter materials alternatingly, as mentioned above, resulting in half of the 22 quartz filter samples being collected on the PSAP S/N 100, and the other half on the PSAP S/N 90. Another custom-built 1-wavelength PSAP ($\lambda$=526 nm; Krecl et al., 2007) used in Svensson et al (2018), was also utilized in for transmittance analysis of all the filters after their production in the airborne- and liquid experiments.

140

### 2.2.2 EC measurements

Punches (typically with an area of 0.64 cm$^2$) taken from the quartz filters were determined for their OC and EC content with a Sunset Laboratory OCEC-analyzer (Birch and Cary, 1996), using the EUSAAR_2 protocol (Cavalli et al., 2010). The analysis procedure is based on step-wise increases in temperature in a helium atmosphere for the first stage, during which OC is detected with a flame ionization detector. The second phase of the analysis consists of introducing oxygen into the temperature increases and the detection of EC. Pyrolysis of OC during the first phase is monitored by a continuous laser transmittance measurement. Once the transmittance has reached the initial value for the filter in the second phase, a separation split-point between OC and EC is established.

### 2.2.3 Size distribution measurements

During the airborne experiments a Grimm optical particle counter (OPC, 1.108) was used as a portable aerosol spectrometer for particle size distributions. The OPC have been factory-calibrated with PSL spheres that are white. Their scattering cross section is larger than that of BC particles which leads to underestimation of particle diameter. We did not find published Grimm 1.108 calibrations with BC particles in the literature, thus we approximated the effect. By using the cross sections modeled by Rosenberg et al. (2012) we estimate that the diameters presented by the OPC are possibly lower by a factor of 2. In Figure 2 we present both the original size distributions and those calculated by multiplying the diameters by 2.

## 2.3 Data processing

Calculations are presented in a step-by-step procedure below. Loading corrections are routinely applied to filter-based measurements of light absorption by atmospheric aerosol, but, for measurements of absorption by melted and filtered snow samples it is not. In the former, absorption is calculated from the product of a loading correction and the rate of change of transmittance, whereas in the latter the absorption is generally calculated simply from the transmittance of the filter only. We therefore show the equivalence of the two methods and that the loading corrections can and should be applied also to melted and filtered snow samples. First, we present a generally use equation for calculating absorption by aerosols, then how the multiple scattering correction factor $C_{ref}$ appears in the equations, followed by how we determined it for the quartz filters. The numerical values of two published loading corrections are given as clearly as possible to save the reader from looking for constants from the literature. Finally, we show the equivalence of calculating the mass absorption coefficients from airborne aerosol and filtered snow samples.

A further note on data processing is important. The single-scattering albedo, $\omega_o$, i.e. the ratio of
scattering and extinction coefficient, is a measure of the darkness of aerosols: for purely scattering
aerosols $\omega_o = 1$. For freshly-generated pure BC, it has been measured to be $\sim 0.2 \pm 0.1$ (Bond et al. 2013).
When pure BC particles get coated with some light-scattering material $\omega_o$ increases so that far from the
sources it is typically larger than 0.9 (e.g., Delene and Ogren, 2002). However, $\omega_o$ varies also with
particle size even for pure BC, in a way that it increases with increasing particle size as can be shown
by the simple Mie calculations in Fig 2b. Both the coating and particle size have consequences for the
analysis of BC in snow by filter-based absorption measurements. The coating of BC particles typically
consists of some water-soluble material such as sulfates, nitrates and organics. The size of BC particles
in snow has been shown to vary in a large range from $\sim 0.1$ µm to $> 2$ µm (e.g., Schwarz et al., 2013).
On the other hand, the estimation of absorption from filter-based attenuation measurements is affected
also by scattering aerosol and therefore by $\omega_o$ (e.g., Arnott et al., 2005; Virkkula et al., 2005; Collaud
Coen et al, 2010). Now, since we do not know the $\omega_o$ of the particles and we will apply the algorithm
presented by Virkkula (2010) we will repeat the calculations with four different $\omega_o$ values. We use the
size distribution measurements for estimating the size and the Mie modeling for estimating a realistic
range of $\omega_o$ for the calculations.

### 2.3.1   Calculation of absorption in aerosols

The PSAP has been calibrated with the standard filter material Pallflex E70-2075W by Bond et al.
(1999; here referred to as B1999) and Virkkula et al. (2005). Ogren (2010; here O2010) presented an
adjustment to the Bond et al. (1999) calibration, while Virkkula (2010; here V2010) updated the
Virkkula et al. (2005) calibration. In all of these the absorption coefficient is calculated as
$$\sigma_{ap} = f(Tr_t)\frac{A}{Q\Delta t}\ln\left(\frac{Tr_{t-\Delta t}}{Tr_t}\right) - s\sigma_{sp} \tag{1}$$

where $f(Tr_t)$ is the loading correction function that depends on the transmittance $Tr_t = I_t/I_0$ where $I_t$ is
the light intensity transmitted through the filter at time t, $I_0$ the light intensity transmitted through a
clean filter at time t = 0, A the spot area, Q the flow rate, and s the fraction of the scattering coefficient
$\sigma_{sp}$ that gets interpreted as absorption and gets usually called the apparent absorption and should be
subtracted from the uncorrected absorption or be treated as presented by Müller et al. (2014). If apparent
absorption can be considered negligible, equation 1 becomes
$$\sigma_{ap} = f(Tr_t)\frac{A}{Q\Delta t}\ln\left(\frac{Tr_{t-\Delta t}}{Tr_t}\right) \tag{2}$$

In the present work, this approach was adapted for two reasons: 1) $\sigma_{sp}$ was not measured during the
calibration experiment and 2) the aerosol used in the experiment was very dark (soot from oil-based
burning), thus the apparent absorption could be considered negligible.

The loading correction function f(Tr) can be further rewritten as $f(Tr) = g(Tr)/C_{ref}$ where $C_{ref}$ is the
multiple scattering correction factor and $g(Tr)$ at $Tr = 1$ a loading correction function that equals one at
$Tr = 1$ and increases when the filter gets darker, i.e., when $Tr < 1$.

$$\sigma_{ap} = \frac{1}{C_{ref}} g(Tr_t) \frac{A}{Q\Delta t} \ln\left(\frac{Tr_{t-\Delta t}}{Tr_t}\right) \qquad (3)$$

If there is only one time step $t = \Delta t$ and before sampling $Tr = 1$ then $Tr_{t-\Delta t} = Tr_{t=0} = 1$ and

$$\sigma_{ap} = \frac{1}{C_{ref}} g(Tr_t) \frac{A}{V_t} \ln\left(\frac{1}{Tr_t}\right) = \frac{1}{C_{ref}} g(Tr_t)\sigma_0 \qquad (4)$$

where $V_t$ is the air volume drawn through the filter since the start of sampling at time t. The assumption
of only one time step means (4) presents the absorption coefficient since the start of sampling on the
filter. According to the Bouguer-Lambert-Beer law light intensity decreases exponentially as a function
of the optical depth $\tau$

$$I_t = I_0 e^{-\tau}$$

$$\Leftrightarrow \tau = \ln\left(\frac{I_0}{I_t}\right) = \ln\left(\frac{1}{Tr_t}\right) \qquad (5)$$
This is relevant especially in the present study since the purpose is to improve estimation of absorption
in filtered snow samples. In the analysis of a snow sample there is only one "time step": $I_0$ is the intensity
of light transmitted through a clean filter and $I_t$ the intensity of light transmitted through a filter through
which the melted snow sample has been filtered. Here the airborne data were also treated in a similar
way: for each time step absorption was calculated from (4) since the start of sampling on the filter.

### 2.3.2 Calculation of $C_{ref}$ of quartz filters

~~Comparison of the $\sigma_{ap}$(quartz) (= $\sigma_{ap}$(Q)) and $\sigma_{ap}$(Pallflex) (=$\sigma_{ap}$(P)) and keeping the published PSAP~~
~~calibration functions (B1999, O2010, and V2010) as standards for $\sigma_{ap}$(P) we derive $C_{ref}$ for the quartz~~
~~filter by the following reasoning. If the same function $g(Tr)$ is used for calculating both $\sigma_{ap}$(Q) and~~
~~$\sigma_{ap}$(P) and especially if the same $C_{ref} = C_{ref,P}$ of the Pallflex filter is used for both filter materials the~~
~~ratio of the absorption coefficients at time t is~~

$$\frac{\sigma_{ap}(Q, C_{ref,P})}{\sigma_{ap}(P, C_{ref,P})} = \frac{\frac{1}{C_{ref,P}} g(Tr_Q) \frac{A}{V_t} \tau_Q}{\sigma_{ap}(P, C_{ref,P})} \qquad (6)$$

~~If this ratio $\neq 1$ and if it is assumed that this is only due to using the same $C_{ref}$ for both filter materials,~~
~~in other words if using filter-material-dependent $C_{ref}$ yielded equal absorption $\sigma_{ap}(Q, C_{ref,Q}) = \sigma_{ap}(P, C_{ref,P})$~~
~~then~~
$$\frac{\sigma_{ap}(Q,C_{ref,P})}{\sigma_{ap}(P,C_{ref,P})} = \frac{\frac{1}{C_{ref,P}}g(Tr_Q)\frac{A}{V_t}\tau_Q}{\sigma_{ap}(P,C_{ref,P})} = \frac{\frac{C_{ref,Q}}{C_{ref,P}}\frac{1}{C_{ref,Q}}g(Tr_Q)\frac{A}{V_t}\tau_Q}{\sigma_{ap}(P,C_{ref,P})} = \frac{\frac{C_{ref,Q}}{C_{ref,P}}\sigma_{ap}(Q,C_{ref,Q})}{\sigma_{ap}(P,C_{ref,P})} = \frac{C_{ref,Q}}{C_{ref,P}}$$

$$\Leftrightarrow C_{ref,Q} = \frac{\sigma_{ap}(Q,C_{ref,P})}{\sigma_{ap}(P,C_{ref,P})}C_{ref,P} \tag{7}$$

~~which means $C_{ref,Q}$ is obtained simply by multiplying $C_{ref,P}$ with the observed ratio of the absorption~~
~~coefficients.~~
If we assume that the difference of the absorption coefficients of the PSAPs using the quartz and Pallflex
filters, $\sigma_{ap}(Q)$ and $\sigma_{ap}(P)$, respectively, is due to the multiple scattering correction factors of the two
materials only we can calculate
$$C_{ref,Q} = \frac{\sigma_{ap}(Q)}{\sigma_{ap}(P)}C_{ref,P} \tag{6}$$

where $C_{ref,Q}$ and $C_{ref,P}$ are the multiple scattering correction factors of the quartz and Pallflex filters,
respectively. However, this is an approximation only since the difference of $\sigma_{ap}(Q)$ and $\sigma_{ap}(P)$ is also
due to the different transmittances $Tr_Q$ and $Tr_P$ of the two filter materials at each time step and
consequently different values of the loading correction. However, below we will use (6) for the
estimation of $C_{ref,Q}$.

The $C_{ref,P}$ values for Pallflex E70-2075W filter were calculated here from two published calibration
experiments. The loading correction function of B1999 (with the O2010 adjustment) can be
reformulated as
$$f(Tr) = \frac{1}{1.5557 \cdot Tr + 1.0227} \tag{$\not{8}$7}$$

This can be further rewritten as
$$f(Tr) = \frac{1}{C_{ref}}g(Tr) = \frac{1}{2.5784}\frac{1}{0.6034 \cdot Tr + 0.3966} \tag{$\not{9}$8}$$

where $C_{ref} = 2.5784$. Similarly, the V2010 loading correction can be rewritten as
$$f(Tr) = \left(k_0 + k_1(h_0 + h_1\omega_0)\ln(Tr)\right) = k_0\left(1 + \frac{k_1}{k_0}(h_0 + h_1\omega_0)\ln(Tr)\right)$$
$$= \frac{1}{C_{ref}}g(Tr) = \frac{1}{C_{ref}}\left(1 + \frac{k_1}{k_0}(h_0 + h_1\omega_0)\ln(Tr)\right) \tag{$\not{10}$9}$$

where $h_0$, $h_1$, $k_0$, and $k_1$ are the constants presented in Table 1 in V2010 and the single-scattering albedo
$\omega_o = \sigma_{sp}/(\sigma_{sp}+\sigma_{ap})$. For the three wavelengths (10) becomes
$$f_{467}(Tr_{467}) = \frac{1}{2.653}\left(1 - 1.698(1.16 - 0.63 \cdot \omega_0)\ln(Tr_b)\right)$$

$$(1\not{1}0)$$

$$f_{530}(Tr_{530}) = \frac{1}{2.793}\left(1 - 1.788(1.17 - 0.71 \cdot \omega_0)\ln(Tr_g)\right)$$

(12~~1~~)
$$f_{660}(Tr_{660}) = \frac{1}{2.841}\left(1 - 1.915(1.14 - 0.72 \cdot \omega_0)\ln(Tr_r)\right)$$

(13~~2~~)
with $C_{ref,467} = 2.653$, $C_{ref,530} = 2.793$, and $C_{ref,660} = 2.841$.
When $C_{ref}$ has been determined it is assumed that $g(Tr)$ is the same for both filter materials.

### 2.3.3 Calculation of mass absorption coefficient (MAC)

If $m_{EC}$ is the mass of EC in the filter (corresponding to the spot area) through which the air volume of
$V_t$ has flown the average mass concentration of EC in aerosol in the air volume is $c_{EC,aerosol} = m_{EC}/V_t$. If
$\sigma_{ap}$ is the absorption coefficient calculated from (4), the mass absorption coefficient (MAC) can be
calculated from
$$MAC = \frac{\sigma_{ap}}{c_{EC,aerosol}} = \frac{\frac{1}{C_{ref}}g(Tr_t)\frac{A}{V_t}\tau}{\frac{m_{EC}}{V_t}} = \frac{\frac{1}{C_{ref}}g(Tr_t)A\tau}{m_{EC}} = \frac{\frac{1}{C_{ref}}g(Tr_t)\tau}{\frac{m_{EC}}{A}} = \frac{f(Tr_t)\tau}{m_{EC}/A}$$
(14~~3~~)

This applies for aerosol but also for the snow samples since the analysis of EC mass in a filter yields
the mass surface density $m_{EC}/A$ in where $m_{EC}$ is the mass of EC in the analyzed filter spot that has the
area A. In Svensson et al. (2018) we calculated apparent MAC values of EC in snow samples simply
from $MAC = \tau/(m_{EC})$ without applying additional corrections for filter loading, neither enhanced
absorption by the filter medium, nor light scattering particles. Assuming that only loading and filter
effects apply in the experiments presented here, the apparent MAC values presented were adjusted by
using $f(Tr,Q) = g(Tr)/C_{refQ}$~~(Q)~~.

## 3 Results and discussion

### 3.1 Airborne aerosol experiment

Through our 22 airborne aerosol samples, we aimed at getting a range of transmittances and EC
concentrations in the filters for the regression analysis. The original goal was to control the final
transmittances by the length of the sampling time, however, this was not always successful (as noted in
Table 1). Without dilution the aerosol concentration in the mixing chamber was very high with
attenuation coefficients $\sigma_0$ in the range of ~60000 – ~90000 Mm$^{-1}$ (see samples 1 and 2, Table 1).
Therefore we added a dilution valve (V1) and a HEPA filter (Fig. 1) after the first couple of experiment
runs, and had variations in the sample air to clean filtered air ratio, which lead to lower $\sigma_0$ in the range
of ~1000 – ~30000 Mm$^{-1}$. The system was not always stable, resulting in different measured
concentrations for similar sampling times.

### 3.1.1 Particle size distribution

The average size distribution measured with the Grimm 1.108 OPC shows that most particles larger
than 1 µm (Fig. 2a) were efficiently removed from the air stream with the pre-separator (Fig. 1). This
is uncertain, however, since the OPC has been calibrated with white PSL spheres (as discussed in 2.2.3).
Another important point is that the lower limit of the sizes the OPC measured was 300 nm, and is
probably even higher due to the above-mentioned calibration error. The particle number size
distribution, nevertheless, suggests that there were large numbers of BC particles smaller than the OPC
detects since the particle number concentration increases sharply with decreasing particle diameter (Fig.
2a).

The mass absorption and scattering coefficients, MAC and MSC, respectively, and single-scattering
albedo $\omega_o$ of single BC particles at $\lambda = 530$ nm were modeled with the Mie code of Barber and Hill
(1990) and the complex refractive index of 1.85 - 0.71i and a particle density of 1.7 g cm$^{-3}$. Comparison
of single-particle $\omega_o$ size distribution (Fig. 2b) with the particle number size distribution (Fig. 2a)
suggests $\omega_o$ varied in the range of ~0.3 – 0.5. Modeling for the size distribution measured with the OPC
yielded $\omega_o \approx 0.51$ and 0.54 when using the original OPC diameters and the diameters multiplied by 2,
respectively. These $\omega_o$ values can be considered as upper estimates considering that a large fraction of
small particles were undetected. However, to take the $\omega_o$ uncertainty into account we calculated all
V2010-related values by using four $\omega_o$ values: 0.3, 0.4, 0.5, and 0.6.

### 3.1.2 Comparison between custom built and commercial PSAPs

The optical depths presented in Svensson et al. (2018) were measured with the custom-made PSAP of
Stockholm University at $\lambda = 526$ nm, which is slightly different than the commercial Radiance Research
PSAP ($\lambda = 530$ nm). Therefore, before applying the corrections (determined in section 3.1.3 below) we
examined whether the transmittances measured with these two PSAPs agree. Transmittances of all
Pallflex and quartz filters were measured with both instruments. The resulting scatter plot (Fig. 3) shows
that the agreement is excellent between the PSAPs, thus we concluded that the corrections established
in this paper could be applied to the results presented by Svensson et al. (2018).

### 3.1.3    Estimation of the multiple-scattering correction factor $C_{ref}$ for the quartz filter

Optical depths ($\tau$) for both the Pallflex and quartz filters, $\tau(P)$ and $\tau(Q)$, respectively, were calculated from (5) at a 1-second time resolution. The $\tau(Q)$-to-$\tau(P)$ ratios – here the $\tau$ ratio – got a wide range of values at 1-second time resolution but most of them were > 1: 99.6 % of $\tau(Q)/\tau(P) > 1$ and the average and median ratios were 1.21 and 1.16, respectively. To study how the $\tau$ ratio depends on filter loading the data were classified into transmittance bins of a 0.025 width in the Tr(P) range of 0.3 – 1.0 and the averages and medians were calculated for each bin (shown in Fig. 3̶4). The transmittance dependence of the $\tau$ ratio of individual samples was often controversial: in some samples it decreased from the beginning, in some samples it increased. We do not have an explanation of this although the high concentrations in the mixing chamber – see the attenuation coefficients $\sigma_0$ in Table 1 – are probably largely the factor behind this observation. However, for all data the average and median $\tau$ ratio depended on the filter transmittance, so that for a fresh clean filter at Tr > 0.9, it was higher than for heavily-loaded filters at Tr < 0.4 (Fig. 3̶4). In addition to the 1-second data the $\tau$ ratio at the end of each sampling period are plotted as a function of transmittance of the Pallflex filter in Fig. 3̶4. For the end values of all samples there was no clear Tr dependence. The most important conclusion in Fig. 4 is that the $\tau$ ratio of the two filter materials depends on the filter transmittance. On the average the ratio decreases with increasing loading even though the same amount of BC is collected on both filters. That suggests that the loading corrections to be applied depend on the filter material and that they do not differ just by a constant factor.

In sample runs 4, 5, 7, 16, 18, 19, and 20 the decrease of Tr was relatively slow and we considered the bin averages and medians calculated from them to be the most suitable to be used for determining $C_{ref}$. Sample 17 was also long, taking more than six minutes. Despite the similar settings used for filling the mixing chamber and the diluter, the $\tau$ ratio was completely different from the rest of the samples (Fig. 3̶4). This outlier was therefore excluded from the analysis.

The two correction algorithms (B1999 and V2010) were next applied to both filter materials and $\sigma_{ap}(Q)$ and $\sigma_{ap}(P)$ (at $\lambda = 530$ nm) were calculated from (4) by using the Tr bin averages and median of $\sigma_0$ and then the ratio of these two, $\sigma_{ap}(Q)/\sigma_{ap}(P)$. When the constants within the correction methods, including the $C_{ref}$, were the same for both filter materials the ratio is close to 1.2 (Fig. 4̶5). As mentioned previously, V2010 depends also on $\omega_o$, and due to the fact that we are unsure of the $\omega_o$ of the aerosol, we present four lines ($\omega_o = 0.3$, $\omega_o = 0.4$, $\omega_o = 0.5$, and $\omega_o = 0.6$) in Fig. 4. The B1999 correction yields a slightly decreasing $\sigma_{ap}(Q)/\sigma_{ap}(P)$, suggesting that only adjusting $C_{ref}$ would not be enough. The V2010 correction does not yield a clear Tr dependence of $\sigma_{ap}(Q)/\sigma_{ap}(P)$ although it has high $\sigma_{ap}(Q)/\sigma_{ap}(P)$ values in the Tr(P) range 0.6 – 0.85. They correspond to the local maxima of the average and median $\tau$

ratio shown in Fig. ~~3~~4. Nevertheless, there is not enough data in this study to robustly test the correction
algorithms. Therefore, all values are calculated with both of them. We calculated next the multiple-
scattering correction factor $C_{ref}$ from (7) by using the Tr(P) bin averages of $\sigma_{ap}(Q)/\sigma_{ap}(P)$. The averages
and standard deviations over the Tr(P,530) range of $1 - 0.3$ and for averaging of all the four single
scattering albedos $\omega_o = 0.3$, $\omega_o = 0.4$, $\omega_o = 0.5$ and $\omega_o = 0.6$ are presented in Table 2. It is worth noting
that $C_{ref} \approx 3.4$ at $\lambda = 530$ nm is close with published values for another commonly used absorption
photometer, the Aethalometer, that uses quartz filters backed with supporting cellulose fibers. For
instance, values around 3.5 were reported by Segura et al. (2014), Zanatta et al. (2016), and Backman
et al. (2017).

## 3.2      Comparison of $\tau$ vs EC of soot mixed in water with airborne particles
The slopes of the optical depths ($f\tau$) vs. EC concentrations, when applying the transmittance-dependent
loading correction f(Tr,Q,V2010, $\omega_o = 0.4$), were different, and depended on how the soot aerosol were
deposited onto the filter (Fig. ~~6~~7a and b). For the airborne aerosol, the slope is $6.4 \pm 0.2$ m$^2$ g$^{-1}$; while
the particles mixed in water (without the ultrasonic treatment) have a slope that is doubled ($12.6 \pm 0.5$
m$^2$ g$^{-1}$). Applying $\omega_o = 0.5$ and $\omega_o = 0.6$ loading corrections, the slopes of the airborne particles are 6.1
$\pm 0.2$ m$^2$ g$^{-1}$ and $5.7 \pm 0.20$ m$^2$ g$^{-1}$, respectively; while the slopes of the particles mixed in water (without
the ultrasonic treatment) are $12.0 \pm 0.4$ m$^2$ g$^{-1}$, and $11.3 \pm 0.4$ m$^2$ g$^{-1}$. The ratios for airborne to liquid
particles are $0.506 \pm 0.026$, $0.507 \pm 0.026$, and $0.508 \pm 0.025$ for the three choices of $\omega_o$ in the
calculation. The difference in slope between the airborne and liquid particles is likely an effect of
penetration depth of the soot particles into the filter media, with the higher slope for liquid particles
reflecting a deeper penetration. Nevertheless, the ratio is named as the water-mixing factor $f_w \approx 0.51 \pm$
0.03. In comparison, using f(Tr,B1999) for the airborne and the water-mixed particles the slopes for
optical depth $f\tau$ vs. EC concentration are $4.33 \pm 0.13$ m$^2$ g$^{-1}$ and $8.31 \pm 0.22$ m$^2$ g$^{-1}$, respectively,
providing a ratio of $f_w \approx 0.52 \pm 0.02$, essentially identical to that obtained from the V2010 correction.

The slope of $f\tau$ vs. EC of the 24 analyzed samples treated in the ultrasonic bath was even higher (Fig.
6a and b), reflecting a probable greater penetration depth of the particles. When f(Tr,Q,V2010) is
calculated with $\omega_o=0.4$, $\omega_o=0.5$ and $\omega_o=0.6$, the slopes of $f\tau$ vs. EC of the particles mixed in water with
the ultrasonic treatment were $18.7 \pm 0.8$ m$^2$ g$^{-1}$, $17.8 \pm 0.8$ m$^2$ g$^{-1}$, and $16.9 \pm 0.7$ m$^2$ g$^{-1}$, respectively.
The average $\pm$ uncertainty of the ratios of the slopes of airborne and water-mixed particles with the
ultrasonic treatment is very stable, $0.34 \pm 0.02$. If we consider this value to be a product of a factor $f_s$
representing the ultrasonic treatment and the above-presented factor $f_w$ we obtain the value $f_s \approx 0.67 \pm$
0.04. When f(Tr,B1999) is used also for the water-mixed and ultrasonic-bath-treated particles the slope
of corrected optical depth $f\tau$ vs. EC concentration is $12.9 \pm 0.4$ m$^2$ g$^{-1}$, with the corresponding $f_s \approx 0.65$
$\pm 0.03$.

The factors are used for multiplying $f(Tr,Q) = g(Tr)/C_{ref}(Q)$, and so another way it can be interpreted is
that they affect the multiple scattering correction

$$f_s f_w f(Tr) = \frac{1}{\dfrac{1}{f_s}\dfrac{1}{f_w}C_{ref}} g(Tr)$$

In other words, $C_{refSW}(Q) = C_{ref}(Q)/(f_w f_s)$ and $C_{refW}(Q) = C_{ref}(Q)/f_w$ for BC particles mixed in water and
filtered through quartz filters with and without an ultrasonic bath, respectively. The values are presented
in Table 2. The uncertainties of $C_{refW}(Q)$ and $C_{refSW}(Q)$ were calculated with a standard error propagation
formula by using the standard deviations of $C_{ref}$s in Table 2 and the above-presented uncertainties of $f_w$
and $f_s$.

To visualize the combined effects of the loading correction functions and the two factors $f_w$ and $f_s$ they
are plotted as a function of $\tau$ in Fig. ~~7~~8. The corresponding transmittances are shown in the secondary
x axis. The range of optical depths of EC in snow presented by Svensson et al. (2018) are also shown
in the figure. It is obvious that the transmittances through those filters were much lower than $Tr = 0.3$
used in the PSAP calibration in V2010 and even more lower than the $Tr = 0.6$ recommended in the
World Meteorological Organization and Global Atmosphere Watch (WMO/GAW, 2011) standard
operating procedures. However, since there is no published calibration for such low transmittances and
high optical depths $\tau$ the approach of extrapolating is the best that can be done. Figure ~~7~~8 also shows
how V2010 and B1999 corrections are close to each other at low $\tau$, but for dark filters at $\tau \approx 2$ there is
a factor of ~2 difference between them.

## 3.3    Implications for field samples
Previously published laboratory and ambient $\tau$ vs. EC regressions in Svensson et al. (2018), were
updated with the above-developed corrections. Svensson et al. (2018) presented linear regressions of
optical depth $\tau$ vs. EC of the same chimney soot we used in the present study, NIST soot (NIST-2975),
and field samples from the Himalaya (India), and Finnish Lapland. ~~However, the optical depths~~
~~presented by Svensson et al. (2018) were measured with a custom-made PSAP of Stockholm University~~
~~at λ = 526 nm, not at λ = 530 nm with the Radiance Research PSAP used in the present study. Therefore,~~
~~before applying the corrections we examined whether the transmittances measured with these two~~
~~PSAPs agree. Transmittances of all Pallflex and quartz filters were measured with both instruments.~~
~~The resulting scatter plot (Fig. 8) shows that the agreement is excellent between the PSAPs, thus we~~
~~concluded that the corrections established in this paper could be done to the results presented by~~
~~Svensson et al. (2018).~~

We multiplied the $\tau$ of the laboratory data of Svensson et al. (2018) with $f_s f_w f(Tr,V2010,\omega_o=0.4,Q)$
since an ultrasonic bath was used also in those experiments. The slopes of the chimney and NIST soot
decreased from ~40 $m^2$ $g^{-1}$ and ~35 $m^2$ $g^{-1}$ to $11.9 \pm 0.9$ $m^2$ $g^{-1}$ and $9.6 \pm 0.6$ $m^2$ $g^{-1}$, respectively (Fig. 9a
and b). In the scatter plot of the chimney soot the two data points with the highest EC concentration of
~0.04 g $m^{-2}$ are possible outliers. When they are discarded from the regression the slope becomes $9.8 \pm$
0.5 $m^2$ $g^{-1}$, which is indicated by the red line in Fig. 9a. This is within the uncertainties and is essentially
the same as for the NIST soot.

These values are now in the order of published MACs, but for chimney and NIST soot still considerably
larger than the $6.4 \pm 0.2$ $m^2$ $g^{-1}$ obtained in the present work (section 3.2). The explanation to this
difference is not clear. However, the procedures of processing the chimney soot and the NIST soot were
not exactly identical to the one we used in the present work. Svensson et al. (2018) mixed both types of
soot manually in MQ water, added some ethanol in the solution and mixed samples with variable
amounts of MQ water before the ultrasonic mixing. In the present work instead, we blew the aerosol
through a virtual impactor into the MQ water, took samples of this solution and diluted the samples
before the mixing in the ultrasonic bath. The two major differences are the use of the size separation in
the present work and the use of ethanol by Svensson et al. (2018), with the explanation being due to
those.

For the re-evaluation of the field data presented by of Svensson et al. (2018) we multiplied the $\tau$ with
$f_w f(Tr,V2010,\omega_o=0.4,Q)$ since the field snow samples were melted and then filtered through the quartz
filters. The slopes of the field samples from the Indian Himalaya and from Finnish Lapland decreased
from $17.1 \pm 0.8$ $m^2$ $g^{-1}$ and $21.5 \pm 0.8$ $m^2$ $g^{-1}$ to $7.5 \pm 0.4$ $m^2$ $g^{-1}$ and $9.8 \pm 0.5$ $m^2$ $g^{-1}$, respectively (Fig
9c and 9d). All slopes above are in the range of published MAC of BC. For instance, Quinn and Bates
(2005) obtained MAC values ranging from 6 to 20 $m^2$ $g^{-1}$, Bond and Bergstrom (2006) and Bond et al.
(2013) reviewed several articles and according to them the MAC of freshly-generated BC is
approximately $7.5 \pm 1.2$ $m^2$ $g^{-1}$ at $\lambda = 550$ nm.~~and Quinn and Bates (2005) obtained MAC values ranging~~
~~from 6 to 20 $m^2$ $g^{-1}$.~~

## 4    Conclusions

Through the airborne laboratory experiments conducted in this study we determined that the multiple
scattering effect is enhanced by about 20% with micro quartz filters compared to Pallflex filters. In
terms of the multiple-scattering correction factor, $C_{ref}$, of the quartz filters, we estimate it to be ~3.4 for
airborne sampled BC. It is worth noting that this is within the range of $C_{ref}$ values published for the
Aethalometer, a very commonly used absorption photometer. The results of the airborne experiments
have also other implications. Atmospheric aerosols are often collected on quartz filters and analyzed
for EC concentration. The same filter samples can also be used for measuring light absorption to derive
the MAC. The analysis showed that if this is done the multiple scattering correction and loading
correction should be taken into account, just as they are in the data processing of online aerosol
absorption photometers.

Mixing BC particles in water and filtering the solution essentially doubled the attenuation of light
compared to airborne generated filters. This is probably explained by the fact that in the liquid phase
and the subsequent filtering the soot particles penetrate deeper into the filter media. Deeper in the filter
substrate, it is more likely that the light absorption effects are enhanced, and that way accounting for
the measured higher optical depth. In the airborne phase the depositional process is most probably
different, with the particulates accumulating in the surface layer of the filter.

When samples were mixed in an ultrasonic bath before filtering through quartz filters the attenuation
was further enhanced. The hypothesis for explaining the effect of the ultrasonic bath is that it possibly
breaks the chain-like structure of BC particles, resulting in smaller BC particles that are able to move
to further depths in the filter matrix. This remains to be confirmed, and can possibly be done with an
electron microscopy. More research on the sampling of BC from melted snow and ice onto filter media
is much needed.

All these effects mean that the absorption data obtained from melted snow samples have high
uncertainties. However, application of the correction functions to earlier published field data from the
Himalaya and Finnish Lapland yielded MAC values of ~7 – 10 $m^2$ $g^{-1}$ at $\lambda = 550$ nm which is in the
range of published MAC of airborne BC aerosol. This gives indirect support for the validity of the PSAP
calibration also for darker filters than used as the limit in atmospheric measurements.

## Acknowledgements

This work has been supported by the Academy of Finland consortium: "Novel Assessment of Black Carbon in the Eurasian Arctic: From Historical Concentrations and Sources to Future Climate Impacts" (NABCEA project number 296302); and the Academy of Finland project: "Absorbing Aerosols and Fate of Indian Glaciers" (AAFIG; project number 268004). J. Svensson further acknowledges personal support from the Maj and Tor Nessling foundation. J. Ström acknowledges support by the Swedish Research Council (VR 2017-03758) "Black carbon particle size distributions from source to sink."

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

Table 1. Main information on aerosol samples taken during the experiment. Sampling time, Tr:
transmittances of Pallflex and quartz filters at $\lambda = 530$ nm at the end of each sample, $\sigma_0$: attenuation
coefficient, calculated without any loading corrections, $\tau(Q)/\tau(P)$: ratio of optical depths of quartz and
Pallflex filters and EC: EC concentration in the quartz filter. The 1-second data from samples denoted
with * were used for deriving $C_{ref}$ of quartz filters. Samples 1 and 2 were taken from the mixing chamber
without any dilution.

| Sample number | Sampling time min | Tr(P) | Tr(Q) | $\sigma_0(P)$ Mm$^{-1}$ | $\sigma_0(Q)$ Mm$^{-1}$ | $\tau(Q)/\tau(P)$ | EC g m$^{-2}$ |
|---|---|---|---|---|---|---|---|
| 1 | 0.55 | 0.314 | 0.279 | 84245 | 92840 | 1.102 | 0.172 |
| 2 | 0.43 | 0.493 | 0.458 | 65284 | 72082 | 1.104 | 0.113 |
| 3 | 1.82 | 0.544 | 0.487 | 13405 | 15842 | 1.182 | 0.094 |
| 4* | 6.7 | 0.543 | 0.509 | 3646 | 4032 | 1.106 | 0.056 |
| 5* | 11.8 | 0.746 | 0.702 | 993 | 1199 | 1.207 | 0.029 |
| 6 | 2.68 | 0.543 | 0.505 | 9103 | 10184 | 1.119 | 0.062 |
| 7* | 12.13 | 0.224 | 0.216 | 4932 | 5052 | 1.024 | 0.195 |
| 8 | 0.6 | 0.609 | 0.592 | 33062 | 34950 | 1.057 | 0.027 |
| 9 | 0.88 | 0.823 | 0.797 | 8821 | 10275 | 1.165 | 0.014 |
| 10 | 0.67 | 0.913 | 0.902 | 5461 | 6188 | 1.133 | 0.016 |
| 11 | 1.38 | 0.931 | 0.923 | 2067 | 2317 | 1.121 | 0.027 |
| 12 | 0.32 | 0.915 | 0.904 | 11221 | 12749 | 1.136 | 0.012 |
| 13 | 0.57 | 0.927 | 0.913 | 5351 | 6425 | 1.201 | 0.009 |
| 14 | 0.65 | 0.814 | 0.781 | 12664 | 15211 | 1.201 | 0.011 |
| 15 | 2.93 | 0.704 | 0.664 | 4786 | 5584 | 1.167 | 0.032 |
| 16* | 11.6 | 0.602 | 0.555 | 1750 | 2030 | 1.16 | 0.029 |
| 17 | 6.12 | 0.5 | 0.415 | 4533 | 5751 | 1.269 | 0.080 |
| 18* | 11.92 | 0.401 | 0.354 | 3067 | 3486 | 1.136 | 0.113 |
| 19* | 10.47 | 0.302 | 0.262 | 4576 | 5119 | 1.119 | 0.147 |
| 20* | 6.97 | 0.402 | 0.367 | 5232 | 5755 | 1.1 | 0.113 |
| 21 | 3.6 | 0.6 | 0.558 | 5676 | 6482 | 1.142 | 0.055 |
| 22 | 2.1 | 0.849 | 0.833 | 3118 | 3480 | 1.116 | 0.017 |


Table 2. Multiple-scattering correction factors of quartz filters. $C_{ref}(Q)$: derived here for airborne BC
particles from published Pallflex filter loading corrections V2010 and O2010. $C_{refW}(Q)$: derived here
for BC particles mixed in water and filtered through quartz filters. $C_{refSW}(Q)$: derived here for BC
particles mixed in water and treated in an ultrasonic bath and filtered through quartz filters.

| | Derived from V2010 | | | Derived from O2010 |
|---|---|---|---|---|
| | 467 nm | 530 nm | 660 nm | same for all $\lambda$ |
| $C_{ref}(Q)$ | $3.23 \pm 0.04$ | $3.41 \pm 0.03$ | $3.48 \pm 0.09$ | $3.08 \pm 0.04$ |
| $C_{refW}(Q)$ | $6.4 \pm 0.3$ | $6.7 \pm 0.3$ | $6.9 \pm 0.4$ | $5.9 \pm 0.2$ |
| $C_{refSW}(Q)$ | $9.5 \pm 0.7$ | $10.0 \pm 0.8$ | $10.2 \pm 0.8$ | $9.1 \pm 0.6$ |



**Figures**

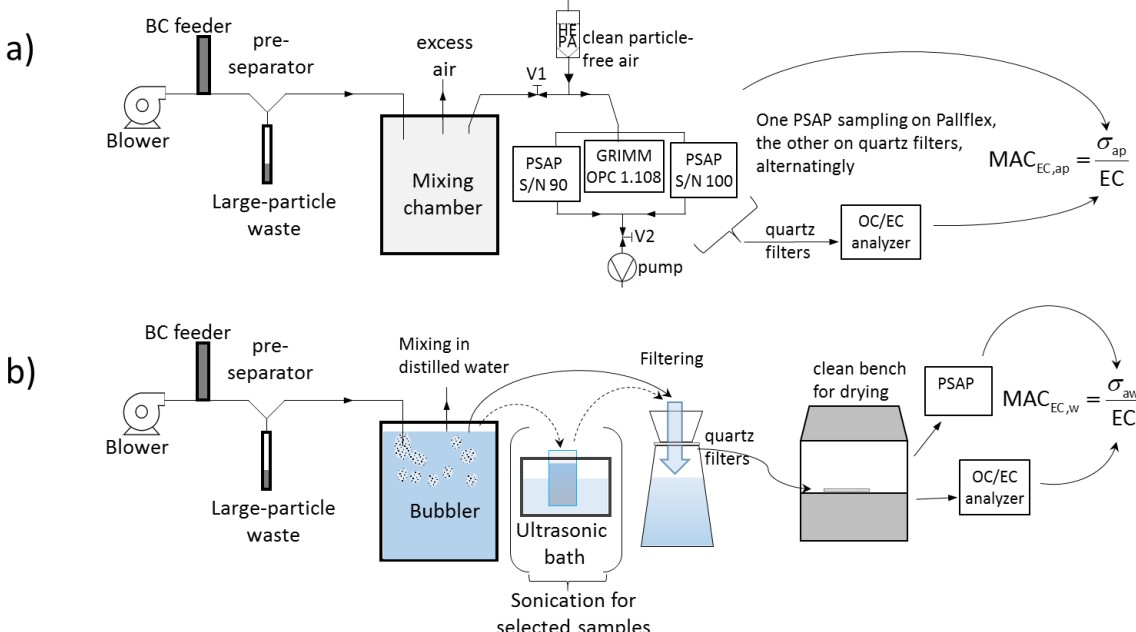


Figure 1. Experimental setup for the airborne (a), and for the liquid (b) procedures.

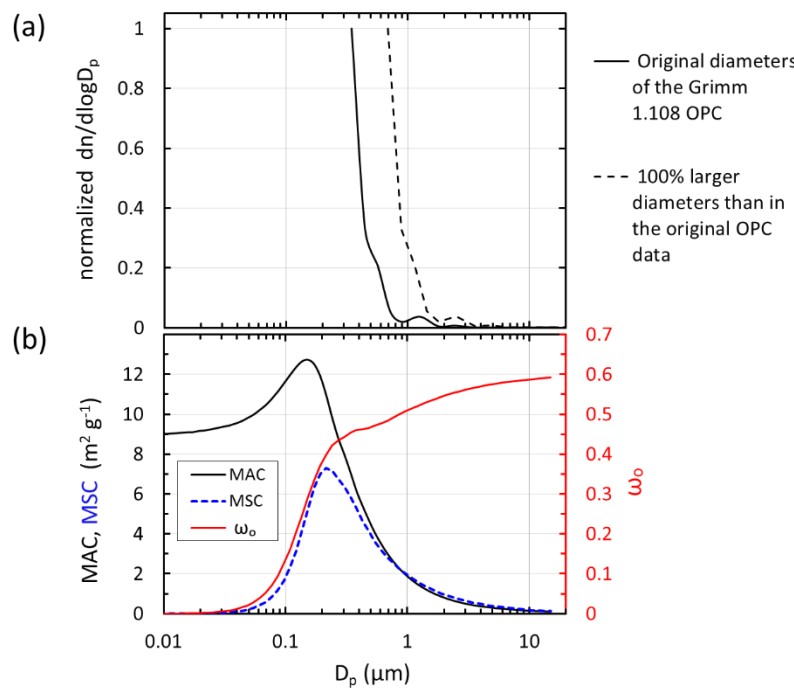


Figure 2. Size-dependent aerosol properties relevant to the experiment. a) Normalized average particle
number size distribution of soot aerosol measurement in the mixing chamber with the Grimm 1.108
OPC. The continuous lines present the size distributions with the original diameters of the OPC and the
dashed lines those assuming that the original diameters were underestimated by a factor of 2. b) Mass
absorption and scattering coefficients, MAC and MSC, respectively, and single-scattering albedo $\omega_o$ of
single BC particles at $\lambda = 530$ nm .

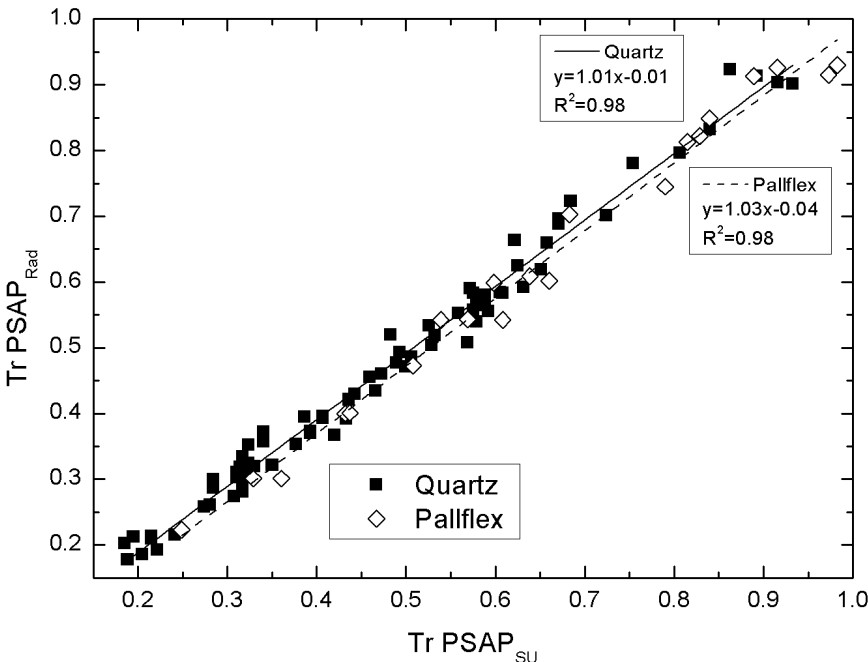


Figure 3. Transmittance for quartz and Pallflex filters measured with PSAP radiance research and the
Stockholm University custom-built PSAP.


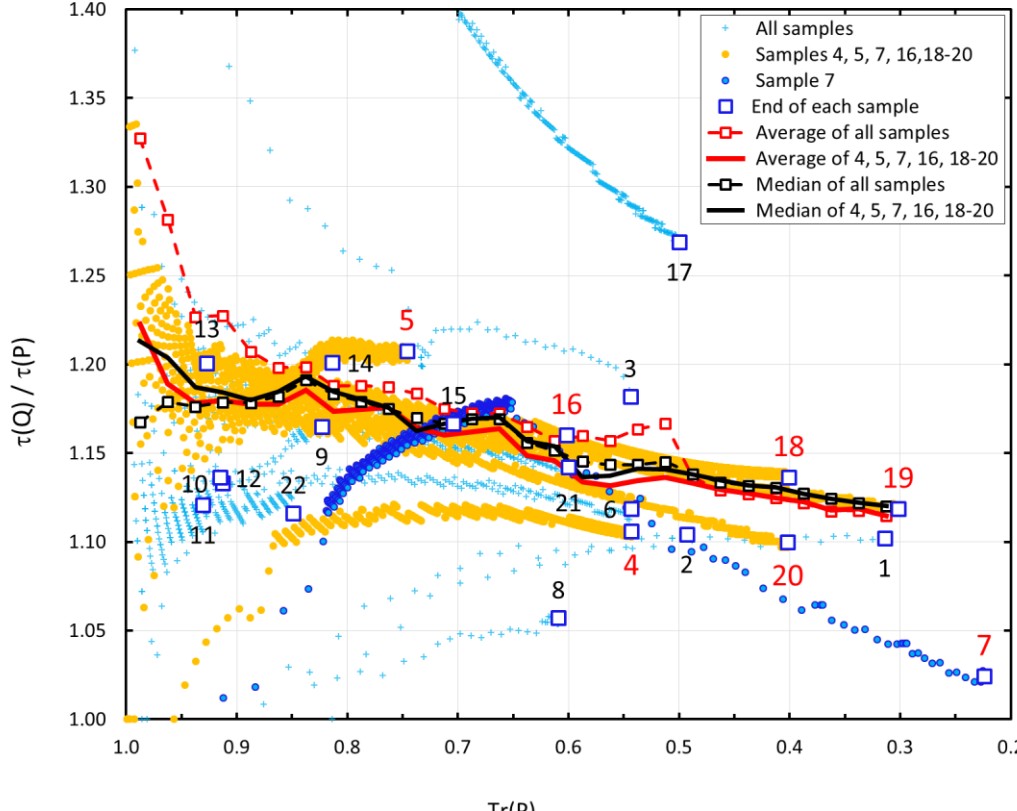


Figure 3̶4. Ratio of non-loading-corrected optical depths ($\tau$ = ln(1/Tr)) of quartz and Pallflex filters,
$\tau$(Q) and $\tau$(P), respectively at $\lambda$ = 530 nm at one second time resolution.. The numbers denote the value
at the end of each sample. The red numbers are associated with those samples that were used for deriving
$C_{ref}$(quartz) in section 3.1.2

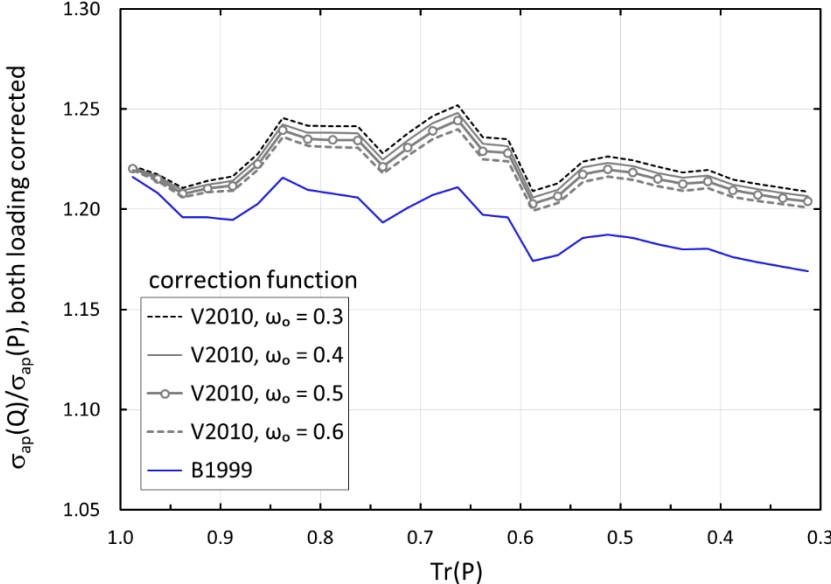


Figure 4̶5. Average $\sigma_{ap}$(quartz)/$\sigma_{ap}$(Pallflex) in 0.025 bins of transmittance of Pallflex filter at $\lambda$ = 530
nm. Both $\sigma_{ap}$(quartz) and $\sigma_{ap}$(Pallflex) were corrected both either according to Bond et al. (1999) with
the Ogren (2010) modification (O2010) and Virkkula (2010) (V2010) using four values for the single-
scattering albedo $\omega_o$.

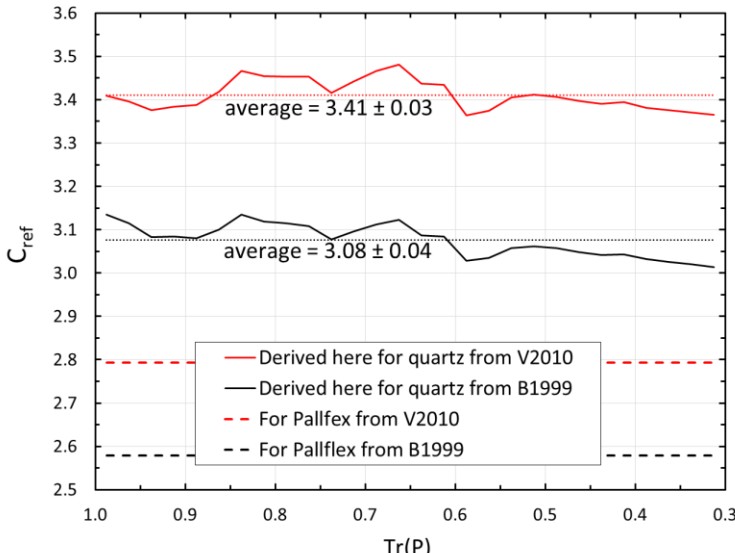


Figure 5̶6. The multiple-scattering correction factor $C_{ref}$ for quartz and Pallflex filters in 0.025 bins of
transmittance of Pallflex filter at $\lambda$ = 530 nm. The straight lines for $C_{ref}$ of O2010 and V2010 are those
shown in Eqs. (9) and (12).

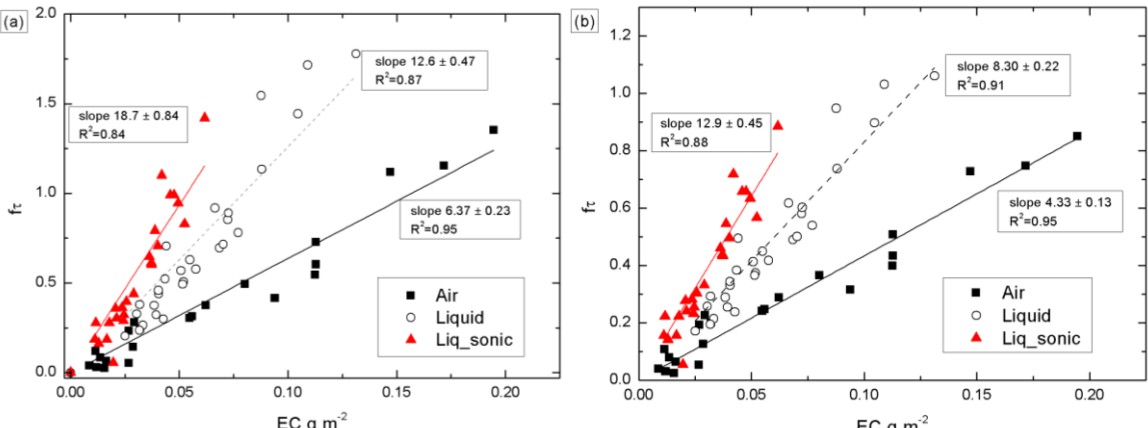


Figure 67. Linear regressions of transmittance-corrected optical depth f$\tau$($\lambda$ = 530 nm) vs. EC of the BC
particles blown into the mixing chamber (Air), into water (Liquid) and blown into water and treated in
the ultrasonic bath (Liq_sonic). The optical depths were corrected with a) the f(Tr,V2010,$\omega_o$=0.4) and
b) f(Tr,Q,O2010). The regressions were calculated by forcing offset to 0.

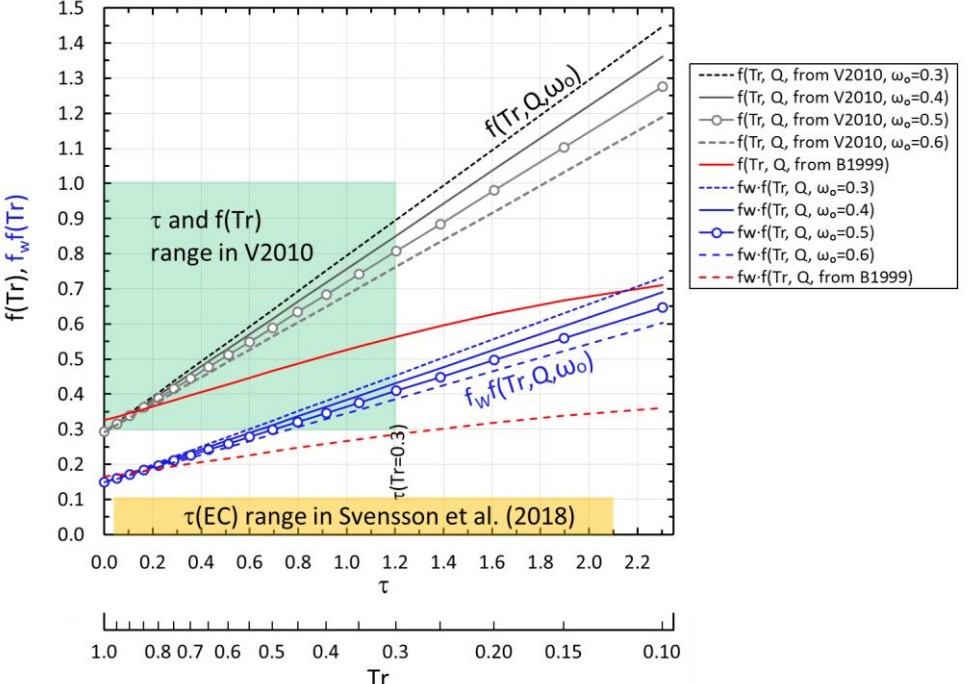


Figure 78. Loading correction functions derived from V2010 and O2010 for airborne BC particles
collected on quartz filters (grey lines, f(Tr,Q,$\omega_o$)) and for BC particles mixed in water and filtered
through similar quartz filters (blue lines, f$_w$f(Tr,Q,$\omega_o$)). The green shadowed area shows the range of
optical depths and f(Tr) of the V2010 Pallflex filter calibration and the yellow shadowed line shows the
range of optical depths of EC in snow presented by Svensson et al. (2018).

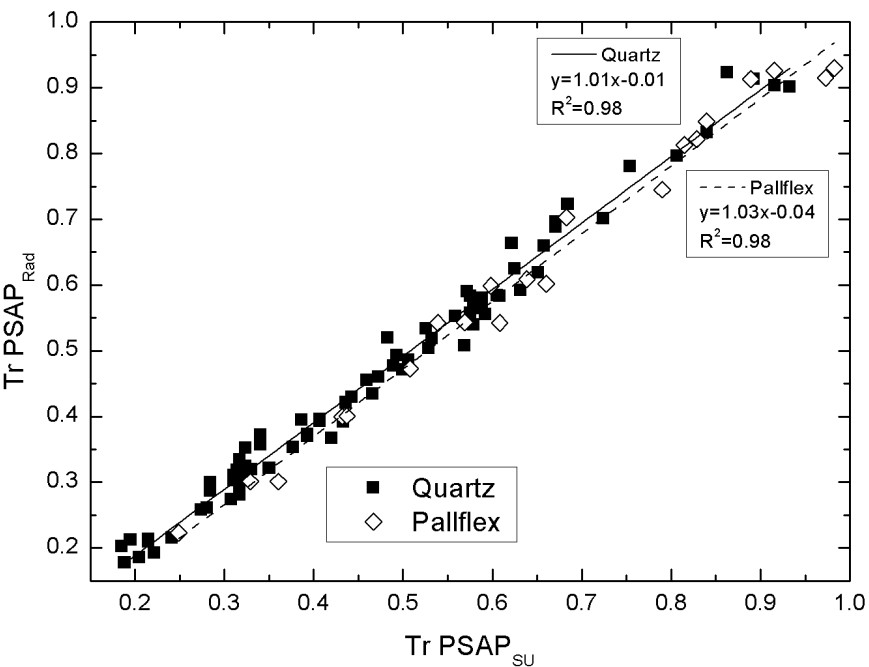


Figure 8. Transmittance for quartz and Pallflex filters measured with PSAP radiance research and the
Stockholm University custom-built PSAP.

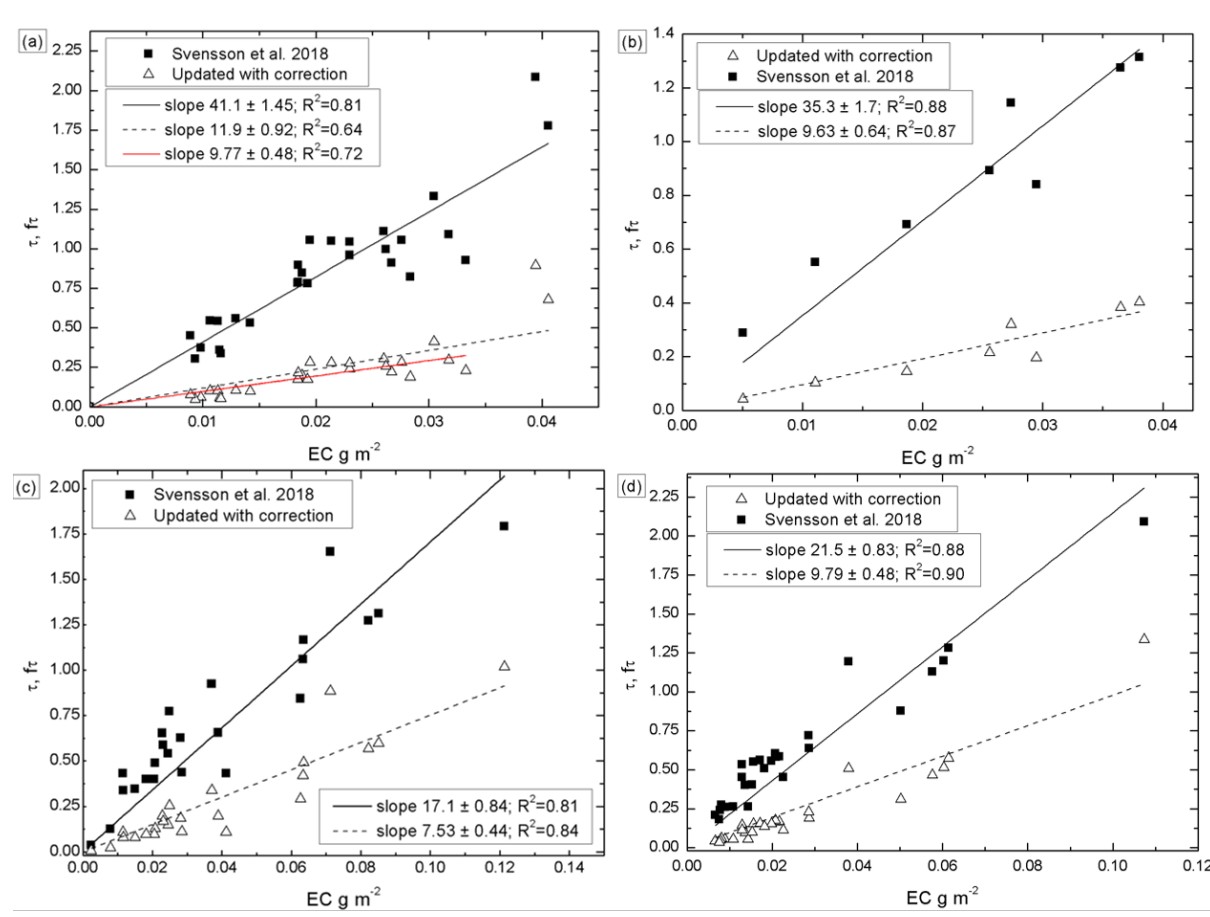



Figure 9. Reanalysis of linear regressions presented by Svensson et al. (2018). a) chimney soot, with
the red line showing the slope with the two points with the highest EC content are excluded, b) NIST
soot, c) field samples from the Indian Himalaya, d) field samples from Finnish Lapland. On the x axis
there is the EC concentration as g m$^{-2}$ and on the y axis the non-corrected and corrected optical depth,
$\tau$ and f$\tau$, respectively.