# Peer review of "Multiple scattering correction factor of quartz filters and the effect of filtering particles mixed in water: implications to analyses of light-absorption in snow samples"

_Atmospheric Measurement Techniques, 2019_

## Referee Comment (RC1) · Anonymous Referee #1 · 18 Jun 2019

The authors address the optical behavior of quartz fiber filters samples with BC. The topic itself is of interest for the AMT community, but also for others such as the climate modelling and climate impact community (e.g. regarding surface snow albedo feedback).

The methods used seem to be scientifically sound, but the presentations lacks my major concern is the structure and the "red line" throughout the manuscript. The manuscript is very hard to read and might gain quality if the whole text would be revised (therefore major revisions) to get it a bit more reader-friendly. Besides, a detailed

discussion of the results is completely missing.

General comments:

Introduction: A short paragraph on the nomenclature should be added. BC as a carbonaceous fraction determined via optical measurement techniques but is strongly related to EC as main light absorbing fraction, but might also include other fractions. These differences should be more highlighted in the introduction.

Section 2.1: The description is a little misleading and hard to follow. A lot of questions arose during reading this section like: - What is the particle size range flowing into the chamber? Cut-off size? - mass ranges of the deposited aerosol - filter material - etc But all these things are getting answered in the subsequent sections. Authors may overthink this kind of structure to help the reader to follow through the rather complex experimental setup. Maybe divide Section 2 in 2.1 airborne sampling procedure and 2.2 liquid sampling procedure.

Section 2.3 Data processing I is very hard to follow the arguments of the authors. What is the message that readers should take or keep for the subsequent sections? Rewrite the section to be more explicit, thereby helping the reader to keep the red line and follow the arguments. Maybe think of reducing the amount of equations, they are not all necessary.

Section 3: Results and discussion A discussion of the results presented is completely missing. E.g. which single-scatterings albedo were found elsewhere? What is the main point authors would like to show in Figure 3? On which filter material and which aerosols were MAC values cited from literature measured?

Conclusion: The main result that the multiple scattering effect is enhanced by 20% with micro quartz filters compared to pallflex filters is not clearly presented.

---

## Referee Comment (RC2) · Anonymous Referee #2 · 16 Jul 2019

Svensson et al. (AMT-2019-142) present a laboratory study of filter-based absorption measurements, designed to provide correction factors for studies on light absorbing particles (LAP) in snow and ice. LAP in snow and ice have previously been measuring by melting and filtering the samples, followed by measurements of transmission through such a sample, from which absorption can be estimated. The major goal of this study is to calibrate such measurements, however, this study did not use representative particles of atmospheric LAP.

First, dust (an extremely important LAP type) was not considered. Furthermore, for the

LAP type focussed on here (black carbon), an inappropriate surrogate material was used. The authors used soot deposits from chimney walls in Helsinki. This is inappropriate, because these soot particles will have coagulated to form new and unique morphologies (larger and more densely agglomerated particles; Dhaubhadel et al., 2006) that do not represent atmospheric black carbon. My criticism is proven by the authors' Figure 2, where substantial numbers of supermicron particles are shown. Such particles are not observed in the atmosphere nor in snow (Schwarz et al., 2012, 2013).

The most robust result of this work is that ultrasonication had a huge effect on their measured calibration factors (the authors described this as "to further mix the soot solutions", in fact the soot suspensions will have either experienced disagglomeration or further agglomeration, depending on the particles and the conditions used. I suspect that disagglomeration will have occurred based on Wang et al., 2012). This proves that particle size was extremely important, meaning that the authors' unrepresentative surrogate black carbon material (chimney soot) has been proven in the authors' own work to have resulted in severely biased and unreliable calibration factors.

The results of this work therefore do not provide a better constraint on transmission-based absorption estimates for filtered meltwater, compared to the reference case of no calibration. In an important sense, they are worse than no calibration, since non-expert readers will assume that "calibrated" measurements are reliable. I would have recommended that the authors use an integrating sphere method (Grenfell et al., 2012) instead of attempting to calibrate a fundamentally limited method. The filter photometer transmittance method is fundamentally a measurement of attenuation and not absorption. The alternative recommendation is to repeat these experiments using dust surrogate particles and freshly-generated soot particles. Unfortunately, the present results will not bring further understanding or clarity to the community and I am obliged to recommend rejection.

**1 Further comments**

Further minor comments:

1. The starting sentence of the introduction is incorrect, soot does not only consist of BC and OC but can also include other materials like sulfates.

2. The statistical treatment of the data was inappropriate. Rather than forcing fits through zero, the authors should either follow the recommendations of Cantrell (2006) and/or calculate mean ratios between the two variables.

3. This paper did not cite or discuss recent important work on determining BC in snow (Schwarz et al., 2012, 2013) nor properly discuss the limitations of the filter-photometer methods. Overall, the literature context of the paper was poor and should be improved. The papers referenced above provide some examples of potential improvements.

**2 References**

Cantrell, C. A.: Technical Note: Review of methods for linear least-squares fitting of data and application to atmospheric chemistry problems, Atmos. Chem. Phys., 8, 5477-5487, https://doi.org/10.5194/acp-8-5477-2008, 2008.

R. Dhaubhadel, F. Pierce, A. Chakrabarti, and C. M. Sorensen. Phys. Rev. E 73, 011404, 2006.

Grenfell, T. C., Doherty, S. J., Clarke, A. D., and Warren, S. G.: Light absorption from particulate impurities in snow and ice determined by spectrophotometric analysis of filters, Appl. Opt., 50, 2037–2048, 2011.

Schwarz, J. P., Doherty, S. J., Li, F., Ruggiero, S. T., Tanner, C. E.Perring, A. E. 2012. Assessing Single Particle Soot Photometer and Integrating Sphere/Integrating Sand-

wich Spectrophotometer Measurement Techniques for Quantifying Black Carbon Concentration in Snow. Atmos. Meas. Tech., 5: 2581–2592.

Schwarz, J. P., Gao, R. S., Perring, A. E., Spackman, J. R. and Fahey, D. W. 2013. Black Carbon Aerosol Size in Snow. Nature Sci. Rep., 3: 1356 doi:10.1038/srep01356

Wang, Mo, Baiqing Xu, Huabiao Zhao, Junji Cao, Daniel Joswiak, Guangjian Wu and Shubiao Lin (2012) The Influence of Dust on Quantitative Measurements of Black Carbon in Ice and Snow when Using a Thermal Optical Method, Aerosol Science and Technology, 46:1, 60-69, DOI: 10.1080/02786826.2011.605815

---

## Author Comment (AC1) · 13 Sep 2019

Our response to the referee are in the supplement

Please also note the supplement to this comment:
https://www.atmos-meas-tech-discuss.net/amt-2019-142/amt-2019-142-AC1-supplement.pdf

---

## Author Comment (AC2) · 13 Sep 2019

Author's responses to the peer-review of Svensson et al. 2019 AMTD "Multiple scattering correction factor of quartz filters and the effect of filtering particles mixed in water: implications to analyses of light-absorption in snow samples."

Here we present point-by-point replies to the referees comments. The referees original comments are presented in black text in this document, whereas our reply is indicated by a *red italic* style. Changes made to the manuscript as a consequence of the referee comments have been inserted at each comment and are indicated by a red color.

On behalf of my coauthors, yours sincerely,

Jonas Svensson

Anonymous Referee #1

The authors address the optical behavior of quartz fiber filters samples with BC. The topic itself is of interest for the AMT community, but also for others such as the climate modelling and climate impact community (e.g. regarding surface snow albedo feedback).

The methods used seem to be scientifically sound, but the presentations lacks my major concern is the structure and the "red line" throughout the manuscript. The manuscript is very hard to read and might gain quality if the whole text would be revised (therefore major revisions) to get it a bit more reader-friendly. Besides, a detailed discussion of the results is completely missing.

*We thank the referee for their comments and review of our paper. As a result of the comments also given by referee #2, we have revised the manuscript and changed some of the structure in the manuscript. In particular, the introduction was changed to guide the reader better on the scope and background of the experiments done here. Overall, the changes made we believe address the comments given by the referee here. See below for our specific answers to the referee's comments.*

General comments: Introduction: A short paragraph on the nomenclature should be added. BC as a carbonaceous fraction determined via optical measurement techniques but is strongly related to EC as main light absorbing fraction, but might also include other fractions. These differences should be more highlighted in the introduction.

*The introduction has been revised in the new version of the manuscript and the nomenclature has been further explained.*

*The introduction paragraphs where the nomenclature (of BC and soot) are touch upon now reads:*

Soot refers to carbonaceous particles formed during the incomplete combustion of hydrocarbon fuels, and includes black carbon (BC) and organic carbon (OC), but can also include other elements, such as sulfates. . As the most light-absorbing aerosol (LAA) by unit per mass, BC is highly efficient in absorbing solar radiation, and is a vital component in Earth's radiative balance (Bond et al., 2013). Once the particles are scavenged from the atmosphere, possibly far from their emission source, BC can reach a snow surface and decrease the snow reflectivity (Warren and Wiscombe, 1980; Flanner et al., 2007).  This will lead to accelerated and increased snowmelt, observed in different snow environments across the globe (see e.g. recent review by Skiles et al., 2018). Perhaps most notably is High Mountain Asia and its extensive cryosphere, where large emission sources of LAA in close proximity is affecting the region's snow and ice (e.g. Xu et al., 2009; Gertler et al., 2016; Zhang et al., 2017).

There are a variety of methods for measuring BC , which is reflected in BC being operationally defined. A common practice is to measure the change in transmission of a filter collecting aerosol. The measured signal (i.e. optical depth of the filter) is thereafter applied with correction factors to generate atmospheric concentrations of so-called equivalent black carbon (eBC) according to the BC nomenclature (Petzold et al., 2013). The correction factors account for: 1) the loading of aerosol on the filter since the detection signal decreases with increased aerosol content; 2) the multiple scattering of light that is enhanced in the filter substrate; 3)

and the enhancement from the deposition of other light scattering aerosol. One instrument used for light absorption measurements is the Particle Soot Absorption Photometer (PSAP), utilizing Pallflex filters. As an alternative for the optical filter analysis of eBC, another approach is to apply the thermal-optical method (TOM), providing organic carbon (OC) and elemental carbon (EC) mass of the aerosol on the filter. With this method, EC refers to the carbon content of carbonaceous matter (Petzold et al., 2013), and can be assumed to be the main light-absorbing element of BC.

Section 2.1: The description is a little misleading and hard to follow. A lot of questions arose during reading this section like: - What is the particle size range flowing into the chamber? Cut-off size? - mass ranges of the deposited aerosol - filter material - etc But all these things are getting answered in the subsequent sections. Authors may overthink this kind of structure to help the reader to follow through the rather complex experimental setup. Maybe divide Section 2 in 2.1 airborne sampling procedure and 2.2 liquid sampling procedure.

*To address this issue we divided section 2.1 into two sections in the revised manuscript, separating the airborne (into 2.1.1) and liquid (into 2.1.2) sampling, as suggested. Further, we added some text in the beginning of section 2, with the aim of guiding the reader of the coming structure of entire section 2.*

*Section 2.1 now reads:*

2.1 Soot aerosol production and sampling

A schematic picture of the experiment is presented in Fig. 1, and the methods used in each step are outlined in the subsections (2.1.1 and 2.1.2) below, as well as the instrumentation used (sections under 2.2). Section 2.3 explains the data processing. The soot used consisted of particles collected by chimney cleaners in Helsinki, Finland, and this particular soot batch is from small-scale oil-based burning. The same soot has been applied in different experiments previously (Peltoniemi et al., 2015; Svensson et al., 2016; Svensson et al., 2018).

2.1.1 Airborne sampling

Soot aerosol were sampled onto filters in an airborne phase and as a part of liquid solution. In the airborne aerosol tests, soot was blown into a cylindrical experimental chamber (0.8 m height × 0.45 m diameter) through a stainless steel tube (25 mm outer diameter) consisting of a y-shaped bend of 130°, creating a size-separation of the aerosol. Essentially a virtual impactor, this set-up allowed the smaller sized particles to continue with the airflow into the chamber, while the larger (and heavier) particles were deposited into a waste pipe through inertial separation (see section 2.2.3 for further description and results in section 3.1.1.).. From the experimental chamber a sample inlet (copper, 6 mm outer diameter) simultaneously fed two PSAPs and a portable aerosol spectrometer (Grimm 1.108). One of the PSAPs had quartz fiber filter punches mounted, while the other had standard PSAP filters installed. This set-up was alternated among the PSAPs in between the experimental runs during the experiment, to have both PSAPs assessed with the different filters. In total, 22 different experimental rounds were made with various amounts of aerosol deposited to the substrates.

2.1.2 Liquid sampling

In the liquid experiments, the same soot batch and procedure were used as above, but the outlet pipe was submerged into a 20 L container filled with deionized, purified Milli-Q (MQ) water. From this liquid solution, different small amounts (between 10-100 mL) were extracted and mixed with additional MQ water to further dilute the sample (to a typical total volume of 400 mL). This was performed to get a range of filters with different EC concentrations and optical depths. The total number of liquid-generated filters was 35. Some selected liquid samples (n=10) were exposed to an ultrasonic bath (for at least 15 mins) prior to filtration . All of the liquid solutions were filtered onto the same quartz filters used in the airborne test, applying the same filtering principles and analysis procedures as used previously (Svensson et al., 2018). Punches from dried filters had their transmittance first measured using a PSAP, followed by EC concentration measurements (TOM). This procedure was also applied to the quartz filters from the airborne experiment.

Section 2.3 Data processing I is very hard to follow the arguments of the authors. What is the message that readers should take or keep for the subsequent sections? Rewrite the section to be more explicit, thereby helping the reader to keep the red line and follow the arguments. Maybe think of reducing the amount of equations, they are not all necessary.

*The section is written in such a way that it proceeds mathematically step by step in detail, and the intension is to be as explicit as possible. Loading corrections are routinely applied to filter-based measurements of light absorption by atmospheric aerosol but not to measurements of absorption by melted and filtered snow samples. In the former absorption is calculated from the product of a loading correction and the rate of change of transmittance whereas in the latter the absorption is generally calculated simply from the transmittance of the filter only. We therefore have to show the equivalence of the two methods and that the loading corrections can and should be applied also to melted and filtered snow samples. We first present the general equations for calculating absorption by aerosols, then how the multiple scattering correction factor $C_{ref}$ appears in the equations and then show how we have determined it for the quartz filters. The numerical values of two published loading corrections are given as clearly as possible to save the reader from looking for constants from the literature. Finally, we show the equivalence of calculating the mass absorption coefficients from airborne aerosol and filtered snow samples.*

*In the revised manuscript, we have removed equation 6 (and the accompanying explanation text) in the beginning of section 2.3.2. For the other parts we believe we need to keep the other equations in order to guide the reader on our reasoning (as explained above).*

*The following text was inserted at the beginning of section 2.3 (as well as another paragraph on single-scattering albedo, as a result of the next comment given on single-scattering albedo) for clarification of the data processing:*

Calculations are presented in a step-by-step procedure below. Loading corrections are routinely applied to filter-based measurements of light absorption by atmospheric aerosol, but, for measurements of absorption by melted and filtered snow samples it is not. In the former, absorption is calculated from the product of a loading correction and the rate of change of transmittance, whereas in the latter the absorption is generally calculated simply from the transmittance of the filter only. We therefore show

the equivalence of the two methods and that the loading corrections can and should be applied also to melted and filtered snow samples. First, we present a generally use equation for calculating absorption by aerosols, then how the multiple scattering correction factor $C_{ref}$ appears in the equations, followed by how we determined it for the quartz filters. The numerical values of two published loading corrections are given as clearly as possible to save the reader from looking for constants from the literature. Finally, we show the equivalence of calculating the mass absorption coefficients from airborne aerosol and filtered snow samples.

A further note on data processing is important. The single-scattering albedo, $\omega_o$, i.e. the ratio of scattering and extinction coefficient, is a measure of the darkness of aerosols: for purely scattering aerosols $\omega_o = 1$. For freshly-generated pure BC, it has been measured to be ~0.2 ± 0.1 (Bond et al. 2013). When pure BC particles get coated with some light-scattering material $\omega_o$ increases so that far from the sources it is typically larger than 0.9 (e.g., Delene and Ogren, 2002). However, $\omega_o$ varies also with particle size even for pure BC, in a way that it increases with increasing particle size as can be shown by the simple Mie calculations in Fig 2b. Both the coating and particle size have consequences for the analysis of BC in snow by filter-based absorption measurements. The coating of BC particles typically consists of some water-soluble material such as sulfates, nitrates and organics. The size of BC particles in snow has been shown to vary in a large range from ~0.1 µm to > 2 µm (e.g., Schwarz et al., 2013). On the other hand, the estimation of absorption from filter-based attenuation measurements is affected also by scattering aerosol and therefore by $\omega_o$ (e.g., Arnott et al., 2005; Virkkula et al., 2005; Collaud Coen et al, 2010). Now, since we do not know the $\omega_o$ of the particles and we will apply the algorithm presented by Virkkula (2010) we will repeat the calculations with four different $\omega_o$ values. We use the size distribution measurements for estimating the size and the Mie modeling for estimating a realistic range of $\omega_o$ for the calculations.

Section 3: Results and discussion A discussion of the results presented is completely missing. E.g. which single-scatterings albedo were found elsewhere? What is the main point authors would like to show in Figure 3? On which filter material and which aerosols were MAC values cited from literature measured?

*As a first point, we believe that the results that we want to present are existing the manuscript. The single-scattering albedo found elsewhere is not a point that we see the need to bring up in the manuscript. We are not presenting any single-scattering ($\omega_o$) values of BC in snow anywhere, $\omega_o$ is needed in the method used in the work. The reviewer's question led us to realize that we had not properly introduced $\omega_o$ and its relevance in the method. Therefore, we added the paragraph about $\omega_o$ to the beginning of the data processing section 2.3. See the additional text in the revised manuscript in the comment above.*

*The main point we want to show with fig. 4 is that the ratio of the optical depth $\tau$ of the two filter materials, quartz and Pallflex is not constant. It depends on the filter transmittance, so that on the average the ratio decreases with increasing loading even though the same amount of BC is collected on both filters. That further suggests that the loading corrections to be applied depend on the filter material and that they do not differ just by a constant factor. Fig. 4 also shows how we choose a selected few runs that seemed most suitable to be used in the calculating the $C_{ref}$ for the quartz filters. This text was added to section 3.1.3.*

*The MAC value of 7.5 ± 1.2 m² g⁻¹ from Bond et al. 2013 is based on a literature survey of 21 publications (Bond and Bergstrom, 2006). It is the average of the measured MACs of freshly generated BC in these papers. It is taken as a benchmark value for MAC in the BC science community, thus a natural value to compare against.*

*For section 3 we have rearranged a section that more accurately reflect the re-written aims structure (as a results of comments given by ref #2), and which we think addresses the structure of section 3 as the comment made here by the referee.*

*In the revised manuscript we inserted a new section 3.1.2 which reads:*

The optical depths presented in Svensson et al. (2018) were measured with the custom-made PSAP of Stockholm University at $\lambda$ = 526 nm, which is slightly different than the commercial Radiance Research PSAP ($\lambda$ = 530 nm). Therefore, before applying the corrections (determined in section 3.1.3 below) we examined whether the transmittances measured with these two PSAPs agree. Transmittances of all Pallflex and quartz filters were measured with both instruments. The resulting scatter plot (Fig. 3) shows that the agreement is excellent between the PSAPs, thus we concluded that the corrections established in this paper could be applied to the results presented by Svensson et al. (2018).

Conclusion: The main result that the multiple scattering effect is enhanced by 20% with micro quartz filters compared to pallflex filters is not clearly presented.

*We find this comment confusing. In the manuscript the first sentence in the conclusion currently reads: "Through the airborne laboratory experiments conducted in this study we determined that the multiple scattering effect is enhanced by about 20% with micro quartz filters compared to Pallflex filters." Hence, we believe this result was already presented in the manuscript's conclusions and have not made any changes.*

Anonymous Referee #2

Svensson et al. (AMT-2019-142) present a laboratory study of filter-based absorption measurements, designed to provide correction factors for studies on light absorbing particles (LAP) in snow and ice. LAP in snow and ice have previously been measuring by melting and filtering the samples, followed by measurements of transmission through such a sample, from which absorption can be estimated. The major goal of this study is to calibrate such measurements, however, this study did not use representative particles of atmospheric LAP.

*We thank the referee for their comments and review of our paper. It appears as if the referee has missed the main point of our paper, as well as several other points. In the revised version of our manuscript we have addressed this issue by restructuring the manuscript, and re-directed the attention to what we want to report in this paper. Please see our detailed answers below to the referee's specific comments.*

First, dust (an extremely important LAP type) was not considered. Furthermore, for the LAP type focused on here (black carbon), an inappropriate surrogate material was used. The authors used soot deposits from chimney walls in Helsinki. This is inappropriate, because these soot particles will have coagulated to form new and unique morphologies (larger and more densely agglomerated particles; Dhaubhadel et al., 2006) that do not represent atmospheric black carbon. My criticism is proven by the authors' Figure 2, where substantial numbers of supermicron particles are shown. Such particles are not observed in the atmosphere nor in snow (Schwarz et al., 2012, 2013).

*It is true that mineral dust is an important LAP type, but the aim of this paper was not to perform tests with mineral dust (or different dust types for that matter). Instead, we wanted to focus on experiments with soot and the particle BC material that we have previously done tests/experiments with (Svensson et al., 2016; 2018), and are currently working with (performing other experiments on snow with this chimney soot). We want to understand this material better and how it influences our measurements (our measurements of BC in snow that is). One crucial part of these experiments, is the fact that we conduct tests with the same BC material that we have previously worked with, in order for our comparisons with the previous results to be valid. In one sense, we can essentially accept any size distribution, as long as it is repeatable.*

*We can agree with the referee that our BC material used is not the ideal proxy for representative atmospheric BC particles, and it is indeed true that supermicron particles of BC are usually not present in the atmosphere. But that is beside the point of the sort of paper we want to report here. It can be further mentioned that in the snow, larger sized BC particles have actually been observed. One of the key results from Schwarz et al. (2013) is that they observed BC particles in the snow that were much larger than those typically seen in the atmosphere (clearly visible in Schwarz et al. 2013 first figure). This has been confirmed by other more recently published papers, e.g. Dong et al., 2018 doi.org/10.5194/tc-12-3877-2018; Schnaiter et al., 2019 doi.org/10.5194/acp-19-10829-2019; Zhang et al., 2017 doi.org/10.1016/j.scitotenv.2017.07.100. Thus, the experiments we have done here with the chimney soot are likely to represent BC particles which could be found in the snowpack. As far as fig. 2 is*

*concerned, we believe that it is important to keep this in the manuscript, as it gives the reader an indication of the aerosol size that were sampled in the experiments.*

*As a result of the above mentioned comments by the referee, we have changed the introduction in the manuscript to more clearly show our aims with this paper.*

*The introduction now reads:*

Soot refers to carbonaceous particles formed during the incomplete combustion of hydrocarbon fuels, and includes black carbon (BC) and organic carbon (OC), but can also include other elements, such as sulfates. . As the most light-absorbing aerosol (LAA) by unit per mass, BC is highly efficient in absorbing solar radiation, and is a vital component in Earth's radiative balance (Bond et al., 2013). Once the particles are scavenged from the atmosphere, possibly far from their emission source, BC can reach a snow surface and decrease the snow reflectivity (Warren and Wiscombe, 1980; Flanner et al., 2007). This will lead to accelerated and increased snowmelt, observed in different snow environments across the globe (see e.g. recent review by Skiles et al., 2018). Perhaps most notably is High Mountain Asia and its extensive cryosphere, where large emission sources of LAA in close proximity is affecting the region's snow and ice (e.g. Xu et al., 2009; Gertler et al., 2016; Zhang et al., 2017).

There are a variety of methods for measuring BC , which is reflected in BC being operationally defined. A common practice is to measure the change in transmission of a filter collecting aerosol. The measured signal (i.e. optical depth of the filter) is thereafter applied with correction factors to generate atmospheric concentrations of so-called equivalent black carbon (eBC) according to the BC nomenclature (Petzold et al., 2013). The correction factors account for: 1) the loading of aerosol on the filter since the detection signal decreases with increased aerosol content; 2) the multiple scattering of light that is enhanced in the filter substrate; 3) and the enhancement from the deposition of other light scattering aerosol. One instrument used for light absorption measurements is the Particle Soot Absorption Photometer (PSAP), utilizing Pallflex filters. As an alternative for the optical filter analysis of eBC, another approach is to apply the thermal-optical method (TOM), providing organic carbon (OC) and elemental carbon (EC) mass of the aerosol on the filter. With this method, EC refers to the carbon content of carbonaceous matter (Petzold et al., 2013), and can be assumed to be the main light-absorbing element of BC. The technique involves a stepwise heating procedure, therefore creating a need to use micro quartz fiber filters. These filters have been used in numerous studies with filtering snow and ice samples, and thereafter analyzed to determine the EC and OC content of the samples (e.g. Hagler et al., 2007; Forsström et al., 2009; Meinander et al., 2013; Ruppel et al., 2014; Zhang et al., 2017). In Svensson et al. (2018), measurements with TOM were combined with an additional transmittance measurement to further  investigate the relative contribution from BC and other LAA particles present in snow samples. The study involved laboratory tests, as well as comparisons to ambient snow samples taken from different environmental settings. One lesson from this study was that the optical properties of absorbing particles on quartz filters must be better understood. In particular when using melted snow samples.

The overarching goal of this paper is to further investigate micro quartz fiber filters optical behavior when sampling BC particles in a liquid (to simulate snow sampling). An advantage of using these filter is

that the sample can be analysed readily using TOM to arrive to an EC concentration on the filter (where MAC values are not needed). Th aim is pursued through a series of laboratory studies . Our approach is to compare the use of quartz fibre filter for air and liquid samples to the much better characterized pallflex type filer used in commercial PSAPs. Hence, we are not intending to determine a universal MAC value, but rather to understand differences in the observations that might be due to the filter substrate or handling of the sample. We do not intend to answer all possible issues with filter sampling, but will concentrate on the difference using the two filter types in air samples, the difference between air and liquid samples with respect to the quartz fibre filter, and finally the potential effect from treating the liquid samples using ultrasound.

The most robust result of this work is that ultrasonication had a huge effect on their measured calibration factors (the authors described this as "to further mix the soot solutions", in fact the soot suspensions will have either experienced disagglomeration or further agglomeration, depending on the particles and the conditions used. I suspect that disagglomeration will have occurred based on Wang et al., 2012). This proves that particle size was extremely important, meaning that the authors' unrepresentative surrogate black carbon material (chimney soot) has been proven in the authors' own work to have resulted in severely biased and unreliable calibration factors.

*On a similar theme as the comments above, we believe that this comment is due to the fact that the point of the paper was missed by the referee. Our approach is to compare the use of quartz fibre filter for air and liquid samples to the much better characterized pallflex type filer used in commercial PSAPs. Hence, we are not intending to determine a universal MAC value, but rather to understand differences in the observations that might be due to the filter substrate or handling of the sample. We do not intend to answer all possible issues with filter sampling, but concentrate on the difference using the two filter types in air samples, the difference between air and liquid samples with respect to the quartz fibre filter, and finally the potential effect from treating the liquid samples using ultrasound.*

*Wang et al. 2012 do not discuss disagglomeration of BC particles, but rather how dust particles are removed from BC particles. From Wang et al (2012): "Carbonaceous matter can be efficiently detached from dust particles by ultrasonic agitation of meltwater samples." Hence, the break-up of dust and carbonaceous matter is clearly different then what has occurred in our study (where no dust was included).*

*The changes made in the introduction (show in the comment above) have addressed this comment.*

The results of this work therefore do not provide a better constraint on transmission based absorption estimates for filtered meltwater, compared to the reference case of no calibration. In an important sense, they are worse than no calibration, since nonexpert readers will assume that "calibrated" measurements are reliable. I would have recommended that the authors use an integrating sphere method (Grenfell et al., 2012) instead of attempting to calibrate a fundamentally limited method. The filter photometer transmittance method is fundamentally a measurement of attenuation and not absorption. The alternative recommendation is to repeat these experiments using dust surrogate

particles and freshly-generated soot particles. Unfortunately, the present results will not bring further understanding or clarity to the community and I am obliged to recommend rejection.

*We are working with this method since it is what we have previously worked with. It is the method readily available to us and we have published papers using it. It is well known that the method we are using has uncertainties and artefats but they have been characterized and analyzed for atmospheric measurements, and partially for BC in meltwater. The method of Grenfell et al. 2012 is certainly also a valid method to measure LAA on filters with. But, once again, that was not the point of this paper and we see no point in switching methods for the purpose of this paper. Performing measurements with dust is not the point of this paper either, nor is it to do it with freshly-generated soot particles (as explained in the comments above). Such a study would be a welcomed addition to the literature, but we truly believe that our paper here is relevant for EC (and BC) in snow studies, as these measurements are unique and have not been done before (not to our knowledge), and will ultimately assist the science community.*

*The work has value also beyond the research of BC in snow. Atmospheric aerosols are often collected on quartz filters and analyzed for EC concentration. The same filter samples can also be used for measuring light absorption to derive the MAC. The analysis showed that if this is done the multiple scattering correction and loading correction should be taken into account, just as they are in the data processing of online aerosol absorption photometers. Our study shows that actually the multiple-scattering correction factor $C_{ref}$ of the quartz filter is very close to that obtained for the Aethalometer, a very commonly-used absorption photometer.*

*In the revised manuscript the point of the paragraph directly above has been included in the first paragraph of the conclusions, which now reads:*

Through the airborne laboratory experiments conducted in this study we determined that the multiple scattering effect is enhanced by about 20% with micro quartz filters compared to Pallflex filters. In terms of the multiple-scattering correction factor, $C_{ref}$, of the quartz filters, we estimate it to be ~3.4 for airborne sampled BC. It is worth noting that this is within the range of $C_{ref}$ values published for the Aethalometer, a very commonly used absorption photometer. The results of the airborne experiments have also other implications. Atmospheric aerosols are often collected on quartz filters and analyzed for EC concentration. The same filter samples can also be used for measuring light absorption to derive the MAC. The analysis showed that if this is done the multiple scattering correction and loading correction should be taken into account, just as they are in the data processing of online aerosol absorption photometers.

1 Further comments Further minor comments: 1. The starting sentence of the introduction is incorrect, soot does not only consist of BC and OC but can also include other materials like sulfates.

*This is correct and the sentence has been changed in the revised manuscript. The starting sentence now reads:*

Soot refers to carbonaceous particles formed during the incomplete combustion of hydrocarbon fuels, and includes black carbon (BC) and organic carbon (OC), but can also include other elements, such as sulfates.

2. The statistical treatment of the data was inappropriate. Rather than forcing fits through zero, the authors should either follow the recommendations of Cantrell (2006) and/or calculate mean ratios between the two variables.

*We believe that the point with our figures (where forcing fits through zero was done) was missed here. The reviewer mistakes the ambition from making the best linear fit to the data with our interest in the slope. Forcing the line to go through origo will most probably make the uncertainty larger for that particular slope, but for the sake of interpretations we relax on the confidence in favor of the interpretation. In figures 7 and 9 we want compare the slope with our previously published values (in Svensson et al., 2018), and how it changes when we apply the corrections. We should therefore use the same standard linear regression as in the original article.*

3. This paper did not cite or discuss recent important work on determining BC in snow (Schwarz et al., 2012, 2013) nor properly discuss the limitations of the filter-photometer methods. Overall, the literature context of the paper was poor and should be improved. The papers referenced above provide some examples of potential improvements.

*Rather than discussing the vast number of papers that now exists on BC in snow (there has been plenty of BC in snow papers since Schwarz et al., 2012, 2013), we attempted to focus on papers that are relevant for the methods and measurements used here. Except for Schwarz et al. (2013), we disagree with the referee and do not think the other references presented by the referee are relevant for our manuscript. A discussion of the filter-photometer methods are presented in the introduction, along with the correction factors (i.e. limitations).*